# Siloxane-Starch-Based Hydrophobic Coating for Multiple Recyclable Cellulosic Materials

**DOI:** 10.3390/ma14174977

**Published:** 2021-08-31

**Authors:** Tomasz Ganicz, Krystyna Rozga-Wijas

**Affiliations:** 1Center of Papermaking and Printing, Lodz University of Technology, 90-924 Lodz, Poland; 2Centre of Molecular and Macromolecular Studies, Polish Academy of Sciences, 90-363 Lodz, Poland; krysia@cbmm.lodz.pl

**Keywords:** siloxane, paper coating, hydrophobicity, starch, wastepaper recycling

## Abstract

The results of the application of a new hydrophobization agent based on a triethoxymethylsilane and standard starch aqueous mixture for mass-produced cellulosic materials—printing paper, paperboard, and sack paper—have been evaluated to examine whether such a mixture can be used in industrial practice. The application of this agent on laboratory sheets prepared in a repetitive recycling process was performed to investigate its influence on the formation and properties of the products, as well as the contamination of circulating water. Measurements of the water contact angle, Cobb tests, and water penetration dynamics (PDA) were performed to test the barrier properties of the resulting materials. The effects of the applied coatings and recycling process on the paper’s tensile strength, tear index, roughness, air permeance, and ISO brightness were studied. Studies have proven that this formulation imparts relatively high surface hydrophobicity to all materials tested (contact angles above 100°) and a significant improvement in barrier properties while maintaining good mechanical and optical performance. The agent also does not interfere with the pulping and re-forming processes during recycling and increases circulation water contamination to an acceptable degree. Attenuated total reflectance Fourier-transform infrared (FT-IR) spectra of the paper samples revealed the presence of a polysiloxane network on the surface.

## 1. Introduction

Chemically unmodified cellulose materials are inherently hydrophilic and do not have good barrier properties against water, which is required for a wide range of applications, from cardboard packaging to the protection of banknotes. As a result of polar interactions and the formation of hydrogen bonds between water molecules and cellulose hydroxyl groups, the contact angle of unmodified cellulosic materials—even on completely smooth surfaces of pure cellulose—ranges, depending on the degree of crystallinity, from 10 to a maximum of 50°, which is well below the practical threshold for non-wetting, which is customarily assumed to be 90° [1]. In addition, natural cellulosic pro-ducts contain larger or smaller amounts of hemicelluloses and lignin, which are even more hydrophilic than cellulose due to the numerous phenolic and acetal groups present in their structure [2]. The lack of barrier properties towards water and steam is in turn due to the porous nature of the structure of cellulosic fibrous materials. With pore sizes of 100–200 μm, they exhibit a strong capillary effect, which is further increased by the low contact angle of their interiors. As a result, water placed on the surface of unmodified paper is quickly absorbed into its interior, while water vapor, having penetrated the pores, is retained on its inner walls, where it then diffuses into the fibers [3].

Despite a large number of publications and patents regarding the hydrophobization of cellulosic materials (see examples: [3,4,5,6]), in practice there are few technologies that really have the potential for industrial application in the production of such mass products as cardboard, coffee cups, or packaging papers because there is a need to reconcile contradictory requirements: low cost of chemicals, the possibility of their application without incurring high investment costs in modifying existing production lines, not creating problems in recycling and, finally, providing barrier properties at a level close to plastic coatings [7]. For niche applications, such as securing banknotes or tissue paper for diagnostic tests, there are less cost and environmental constraints—hence, many recent publications on the hydrophobization of cellulosic products describe complex methods leading to superhydrophobic properties [8,9], combining hydrophobicity with other properties (e.g., bacteriostatic) [10], or providing the ability to selectively control hydrophobicity at specific locations [11].

Among the new hydrophobizing technologies offering the greatest hope for mass applications are those based on modifications of traditional methods using alkenyl derivatives of succinic acid anhydride (ASA) and alkylketene dimers (AKD) by combining them with natural or biodegradable polymers [12], and attempts to adapt silicone agents previously developed for building materials [13,14]. Hydrophobizing mixtures based on reactive organosilicon compounds are particularly attractive for environmental reasons because they decompose to silica, unlike many other formulations that contaminate water with hazardous chemicals [15]. Unfortunately, most of the organosilicon formulations found in the literature to date for hydrophobizing cellulosic materials are either too expensive or too complex for mass application in the paper industry [4]. Most of them are based on reactive aminosilanes [8] or chlorosilanes [9,16] in organic solvents, which rapidly condense with water, releasing harmful by-products into the environment; therefore, their application in the highly humid environment of paper machines seems to be impossible. The most promising agents are those based on alkoxysilanes [13,17]. Their condensation with water results in relatively harmless aliphatic alcohols as side products, and the progress and equilibrium of this reaction can be controlled by pH buffers and the relative concentration of di-, tri-, and mono functional silanes [18]. In addition to the formulation and use of water repellents themselves, an important issue is the effect of their use on paper recycling processes. While many studies can be found in the literature on changes in the structure of cellulose fibers during multiple pulping and paper forming cycles [19,20], as well as general analyses of the environmental aspects of these processes [15,21], the impact of organosilicon agents in this regard has not yet been studied.

The present work continues the study of a new hydrophobizing agent based on an aqueous solution of methyltriethoxysilane stabilized with polyvinyl alcohol and a commercial surfactant. Previous studies have demonstrated the ability to effectively hydrophobize laboratory-made paper sheets based on pure bleached softwood pine kraft pulp (BSK) containing no chemical additives [14]. However, these studies did not answer the question of whether this agent would be effective for mass-produced paper, which contains many additives such as mineral fillers, modified starch, sizing agents, surface modifiers, etc. It was also necessary to check whether the applied coating does not have a negative impact on the possible recycling processes, in terms of the contamination of the circulation water generated during the pulping of wastepaper, the paper forming processes, and the most important properties of the final products.

The research described—the application of an agent based on a mixture of alkoxysilane and starch to the hydrophobization of mass-produced cellulosic materials (printing paper, paperboard, sack paper), and the study of the effect of the application of this agent on the formation and performance of laboratory sheets in multiple recycling processes—was intended to answer these questions.

## 2. Materials and Methods

### 2.1. Chemicals

The following chemicals were used as provided by vendors: methyltriethoxysilane (98%, ABCR GmbH, Karlsruhe, Germany), poly(vinyl alcohol) (PVA) (Mowiol^®^ 8-88, Sigma-Aldrich, St. Louis, MO, USA, molar mass = 6.7 × 10^4^ g/mol, reagent grade), commercial wheat starch (C*Flex 20002, Cargill, Incorporated, Minneapolis, MN, USA, containing 31.16%wt. ± 0.15%wt. amylose, 0.24%wt. ± 0.02%wt. lipids, and 0.29%wt. ± 0.01% wt. proteins), and sodium C12-15 pareth sulfate solution (Sulforokanol L-327, Silikony Polskie Ltd., Nowa Sarzyna, Poland). The water used for the preparation of hydrophobizate was demineralized by reverse osmosis (RO-5, Aquafilter Europe Ltd., Lodz, Poland) from tap water.

Methyltriethoxysilane emulsion in water (SIL, 50%wt.), starch/NaOH sol in water (ST, 4%wt. of starch and 0.2%wt. of NaOH), and final coating mixture were prepared using the method already described in our previous publication [14]. The proportion of SIL to ST mixtures in final coating agent was 1:2.

### 2.2. Cellulosic Materials

#### 2.2.1. Laboratory Handsheets

Laboratory handsheets were obtained using commercial, bleached softwood pine kraft pulp (BSK) with an *α*-cellulose content of 86.6%, Schopper–Riegler value of SR-12, initial moisture content of 93.78%, and a degree of polymerization of 1081. Pulp samples were prepared following procedure described in ISO 5263-1 (2004) standard. Laboratory handsheets were formed according to ISO 5259-2 (2001) standard on Rapid-Köthen (Model 2004, Labor-Meks, Lodz, Poland) apparatus. Before coating, the handsheets had average basis weight of 80 g/m^2^. 

#### 2.2.2. Commercial Paper Sheets

The coatings were also applied on the following typical commercially available papers:White printing paper: Prime Jet 80 (International Paper Poland, Poland);Cardboard testliner: AvantLiner Recycled 85 (Stora Enso Poland, Ostrolęka, Poland);Unbleached Sack Kraft Paper: Optima S 80 (Stora Enso Poland, Ostrolęka, Poland).

AvantLiner and Optima S were provided in 25 × 1 m rolls, from which A4 sheets were cut in random places with longer edge parallel to the paper machine direction. Prime Jet was provided in typical pre-cut A4 500-sheet pack.

### 2.3. Coating Paper Surface

To coat the specimens of papers with the hydrophobizing mixture, an automatic coater (Control Coater, TUL, Lodz, Poland) with standard Mayer rod No. 5 (K-bar) was used. Moving the Mayer rod at a speed of 16 m/s resulted in the formation of a uniform liquid layer of the mixture with thickness of 50 µm. The hydrophobizate was always applied to the top side of the samples. In the case of AvantLiner and Optima S papers, this was the smoother side, which is intentionally on the outside of the final products (boxes or bags). In the case of Prime Jet paper, it was the top side after the sheets were taken out of the packaging in which they were sold; laboratory handsheets were coated on the side that does not come into contact with the wire during forming. The coated samples were dried on laboratory rotary drum dryer (Type 89, Mechanika Praha, Czech Republic) at a temperature of 100 °C ± 2 °C for 2 min. The arithmetic average weights of the handsheets and commercial paper samples before and after coating are listed in Table 1.

### 2.4. Recycling Experiments

#### 2.4.1. Disintegration, Forming, and Coating Cycles of Handsheets

After preparation of first set of handsheets (cycle 1), remaining coated samples after all non-destructive measurements were disintegrated to pulp according to the standard procedure described in ISO 5263-1 (2004) using tap fresh water. Laboratory handsheets were formed again in a Rapid-Köthen apparatus (Model 2004, Labor-Meks, Lodz, Poland) according to ISO 5259-2 (2001), coated again according to the method described in Section 2.3, measured and disintegrated again. The process of forming, coating, measuring, and disintegrating was repeated 3 times (cycles 2–4). Additionally, for comparison, similar cycles of forming, measuring, and disintegrating but without coating were performed for a separate set of reference handsheets, starting from cycle 2. The arithmetic average weights of the handsheets after each cycle are given in Table 2.

#### 2.4.2. Water Contamination Analyses

During each cycle of forming, handsheet samples of post-process water were collected and analyzed: turbidity (NTU units) was measured using a Lovibond TB 211 IR turbidimeter (Tintometer GmbH, Dortmund, Germany) following the ISO 7027 standard. COD (Chemical Oxygen Demand) was determined according to ISO 6060:1989 standard using 1.14541.0001 COD Cell Test Spectroquant kit and the Spectroquant VEGA 400 spectrophotometer (Merck, Germany). pH was measured using S210-Uni pH-meter equipped with InLab Expert Pro electrode (Metler Toledo, Switzerland), calibrated using Mettler Toledo buffers: 2.00, 4.01, 7.00, 9.21 and 11.00. The results of these measurements are given in Table 3.

### 2.5. Mechanical and Surface Properties of Paper

The paper specimens were conditioned following the ISO 187:1990 standard procedure. The properties of the samples were measured in accordance with the following ISO standards: Elmendorf tear resistance (ISO 1974:1990), tensile index and elongation (ISO 1924-2:2008), Bendtsen air permeance (ISO 5636-3:1992), Bendtsen surface roughness (ISO 8791-2:1990), and ISO brightness (ISO 2470-1:2016).

#### 2.5.1. Contact Angle

The contact angles were measured using a PGX goniometer (Testing Machines, Inc., New Castle, DE, USA) by using the TAPPI T 458 (2004) standard static method. The values shown in the tables and graphs are the arithmetic means of 3 measurements of randomly selected samples of each paper type.

#### 2.5.2. Water Penetration Dynamics (PDA)

A penetration dynamics analyzer (PDA) (Module S 05, Emtec Electronic GmbH, Leipzig, Germany) was used to study water penetration dynamics. The strips of the dimensions of 7 cm × 3 cm each were cut from 3 samples of each paper type. Measurements were performed in demineralized water at 20 °C, according to the standard procedure described by the manufacturer. The resulting PDA plots were arithmetically averaged from three measurements for each type of sample.

#### 2.5.3. Water Absorptiveness and Durability

Water absorptiveness studies were performed using Cobb test (ISO 535:2014), conducted for 120 s and 20 h exposure times to water.

#### 2.5.4. Scanning Electron Microscopy (SEM)

SEM images were taken using a JSM-5500 LV (JEOL Ltd., Tokyo, Japan) scanning electron microscope. Standard conditioned (ISO 187) samples of paper were cut into 1 × 1 cm stripes. A nanolayer of gold was applied on their surfaces using a standard vapor deposition process.

#### 2.5.5. Infrared Spectra (IR)

Attenuated Total Reflectance (ATR) FT-IR spectra of the test paper samples were measured on a Jasco 6200 spectrophotometer (Jasco Corp, Tokyo, Japan). Spectra were recorded averaging 64 interferograms with a resolution of 4 cm^−1^. The measurements were repeated for three independent samples of paper, recording the spectrum on top (coated) side of the samples.

## 3. Results and Discussion

### 3.1. Application of Hydrophobizate on Industrially Produced Papers

Our previous study proved that a new hydrophobizate consisting of methyltriethoxysilane, starch, and water led to papers possessing relatively high surface hydrophobic properties (water contact angle over 100°) as well as barrier properties and water resistance [14]. However, that study was limited to highly porous handsheets laboratory-made from pure bleached softwood pine kraft pulp. Industrial paper materials such as sheets for laser printers, cardboard liners, or packaging papers usually contain various additives, which may or may not interfere with alkoxysilanes. Therefore, to examine the usefulness of the new hydrophobization formula, it was applied on three types of industrially produced papers: Prime Jet 80 from International Paper Poland, representing typical printing paper, AvantLiner Recycled 85—typical 100% recycled paper for the outer layer of general-purpose brown cardboard, and Optima S 80—typical unbleached packaging paper, both from Stora Enso Poland, using the method developed for handsheets. The samples represent typical mass-produced machine papers, but to keep the results of experiments comparable, the products with the same basic weight (80–85 g/m^2^) were selected.

The hydrophobizing agent consisted of a stable 50%wt. water emulsion of methyltriethoxysilane (SIL) and 4%wt. wheat starch with 0.2% NaOH sol in water (ST) vigorously mixed in 1:2 SIL:ST volume proportion just before application on paper. The method of application—using an automated coater with a Mayer rod and subsequent drum drying—approximated the industrial processes performed on starch presses and the drying rollers of contemporary paper-forming machines. After applying the coatings, the average weight of the sheets increased by 1.33–2.78% for machine papers and 4.51% for handsheets (Table 1). The main reason for the difference is probably due to the ability of the materials to be penetrated by the hydrophobizing agent. Larger differences in coating weights are seen for papers that have higher water absorption in the 120 s Cobb test (handsheets and Prime Jet 80) and smaller differences for samples with low water absorption (AvantLiner and Optima S).

### 3.2. Hydrophobic Surface and Barrier Properties

The barrier properties of the samples were studied using three methods. The surface hydrophobicity was determined by the measurement of the contact angle of water drops placed on the surface of the paper according to the TAPPI T 458 (2004) standard. Water penetration dynamics were analyzed using the PDA and Cobb methods. Both methods were performed for reference samples of untreated sheets and papers coated by a starch–hydrophobizate mixture with always the same proportions of components (1:2 SIL:ST vol.), which were found to be optimal in a previous study [14].

The results of contact angle measurements are presented in Figure 1. For uncoated handsheets, the static contact angle was unmeasurable as the water drops had been completely absorbed within 15 s. For all other types of papers, the contact angles of untreated samples were in the range of 78.5–90.1°. The application of the hydrophobizing agent increased the angle from 106.8° (Optima S 80) to 112.6° (Prime Jet 80). It is interesting that, except handsheets, the lower the contact angle of the untreated samples, the higher the angle after coating.

Cobb 120 s measurements (Figure 2) provide a basic insight into the barrier properties of the tested materials. It seems that the lower the barrier of uncoated samples, the higher the effect of the coating. It is very high for handsheets (129.5%wt. water intake for uncoated samples vs. 49.6% for coated ones), substantial for Prime Jet 80 and Optima S 80, and negligible for AvantLiner 85, which originally had relatively high barrier properties.

To test the effectiveness of the presented hydrophobization method in prolonged contact with water, extended Cobb tests were carried out, during which the time of contact with water was 20 h (Figure 3). 

For uncoated handsheets and Prime Jet 80 samples, it was impossible to measure the water intake in Cobb 20 h experiments as they lost their physical integrity when exposed to water for such a long time, while their coated counterparts had very good durability. For Optima S 80, the coating resulted in almost a 50% increase in durability, and for AvantLiner 85 the effect was less than 20% but still visible.

The PDA method enables real-time assessment of the penetration of water in which the sample is immersed. The intensity of infrasound passing through the sample from generator to detector depends on its composition. It is much lower for water than for air; therefore, if the material has a surface barrier, there is a peak in the measured signal, which then decreases due to water penetration into the paper.

For samples possessing originally low barrier properties (handsheets and Prime Jet 80), the positive effect of coating is very visible (Figure 4a), while for papers with a naturally higher barrier toward water the effect is less visible, and, especially for Optima S 80, seems to be mainly connected with the surface barrier rather than with mass wetting (Figure 4b).

Overall, the above results lead to the general conclusion that the hydrophobizing agent has a double effect—it increases surface hydrophobicity in roughly the same way regardless of composition and increases wetting barrier properties, especially if they were initially low.

### 3.3. Recycling Cycle Studies

From an environmental point of view, the chemical composition of the hydrophobization agent seems to be relatively safe. Methyltriethoxysilane is a commonly used crosslinking agent for silicone resins in many applications. A 2020 analysis based on Article 48 of the EU REACH Directive classified this substance as a mild skin and respiratory irritant (at high concentrations) and no other hazards were identified [22]. Sodium C12-15 pareth sulfate solution (Sulforocanol L-327) is commonly used as a detergent in products such as shampoos, face washes, car wash formulations, etc. Among many other surfactants, this substance was found to be one of the safest for the environment and humans [23]. The by-products of the condensation of methyltriethoxysilane are only ethanol and possibly its esters with sulfate fatty acids, which are also not a major environmental concern.

However, in the case of cellulosic materials that are supposed to be recyclable, it is important that they do not contain additives that might seriously affect the processing of wastepaper in paper mills. This may happen due to the negative influence on disintegration and forming mechanisms as well as the contamination of circulating water and wastewater. To study this problem, four cycles of forming, coating, and disintegration of handsheets made of pure bleached softwood pine kraft pulp were performed. In each cycle, the hydrophobic, mechanical, and optical properties of obtained handsheets were studied, and basic contamination parameters of post-process water were determined. This was compared with a reference set of handsheets that were recycled in the same way, but without coating.

The average results of water penetration (Cobb 120 s) and hydrophobicity of surface (contact angles) measurements performed for handsheets formed in each cycle are presented in Figure 5. The static contact angles for uncoated samples were unmeasurable as the water drops had been completely absorbed within 15 s.

Within the limits of the statistical error, Cobb 120 s water absorptiveness rates (21 ± 2%) and the contact angles (108 ± 1°) for coated samples did not change with the successive sheet forming cycles. Additionally, there was no substantial change in Cobb 120 s absorptiveness rates for reference samples, which were recycled without being coated. However, there was a substantial decrease in water absorptiveness rates for uncoated samples, prepared from previously coated ones in subsequent forming cycles, from 119.3% for cycle 1 and 95.4% for cycle 4, which suggests that some amount of hydrophobizate accumulates on the surface of the fibers. This corresponds well with the constant increase in the average mass of uncoated samples (Table 2) with the number of forming cycles, despite using fresh water for the disintegration of handsheets, which again is not observed for reference samples.

The wastewater contamination studies (Table 3) suggest that after the 3rd cycle of forming, there is a kind of equilibrium between the part of the hydrophobic substances that pass into the water and the part remaining on the fibers. Moreover, the results for the reference samples that were recycled without coating indicate that in the 1st and 2nd cycle, some of the short cellulose fibers went through the forming wire to the water, while in the following cycles they did not. Thus, the water contamination parameters in cycle 3 and 4 for handsheets coated with hydrophobizate are probably due to its use only. Anyway, the COD (less than 800 mg/dm^3^) and turbidity (less than 30 NTU) of the wastewater are well below typical industrially accepted ranges (COD below 2500 and NTU below 100) for all four cycles for coated samples [24]. However, in the case of possible practical applications of the tested chemicals, the relatively high pH of the whitewater may necessitate an extra neutralization before it is returned to circulation in the paper machine.

### 3.4. IR and SEM Studies

The interactions of alkoxysilanes with paper and starch, examined using ^29^Si solid state NMR and SEM, suggest that alkoxysilanes condensate with each other, producing silicone microparticles, and do not react with the starch or paper surface directly, so there are no covalent bonds Si–O–C with carbon atoms constituting cellulose and starch polymeric chains [14]. To study this further, FT-IR spectra of coated and uncoated samples using attenuated total reflection (ATR) mode were recorded. The study was performed with white printing paper (Prime Jet 80) as having the highest IR reflection rates and thus producing the spectra with the highest quality among the tested types of paper.

In the IR spectra of cellulosic materials coated with formulations based on silanol condensation products, the following bands are key to establishing the presence of silicon-containing substances bound to the surface: 1270 cm^−1^ (Si–CH_3_ stretching vibration), 1127 cm^−1^ (Si–O–Si stretching), 1034 cm^−1^ (Si–O–C stretching), and 770 cm^−1^ (Si–CH_3_ bending) [25]. Of these signals, only the one at 1034 cm^−1^ may be evidence of the covalent bonding of silanol condensation products to the hydroxyl groups of cellulose and starch. The others prove that the organosilicon material has deposited on the paper surface but is not necessarily covalently bound with it [17,26]. This is complicated by the fact that some of these key signals overlap with the cellulose and starch spectra. Nevertheless, the strong reduction of the bands at 1413 and 872 cm^−1^, originating from the semiacetal groups in starch and cellulose, suggests that bond formation has occurred between these substances and the silanol condensation products. Unfortunately, the key, weak signal at 1034 cm^−1^ is completely covered by 1020 cm^−1^ Si–O–Si vibration in the coated sample, and therefore cannot be used to prove covalent bonding between silicate particles and the cellulosic surface. However, there is no doubt that the main substances covering the surface of paper are the condensation products of silanols with themselves, as indicated by the shift of the intense band at 1060–1034 cm^−1^ to 1020 cm^−1^ (Si–O–Si bonds) and the narrow line at 1270 cm^−1^ originating from Si–CH_3_ vibrations (Figure 6) [27].

The Prime Jet 80 paper is already factory-treated with starch and probably the inorganic crystalline substances responsible for the optical whitening effect, which is clearly seen in SEM images (Figure 7a). These substances also remain visible in the image of the sample coated with hydrophobizate. As expected, the image of the reference sample of the handsheet shows only pure fibers forming a structure with open pores.

The main, readily apparent difference in the SEM images of coated samples is the presence on their surface of numerous spherical, randomly scattered objects with average sizes of 2–3 micrometers, which, judging from the IR spectra, consist mainly of silanol condensation products—possibly branched polysiloxane networks (Figure 7b and Figure 8b,d). However, in the case of the Prime Jet 80 sample, the surface coverage with the spherical objects is much denser than for handsheets, despite using the same concentration of siloxanes in the coating mixture. This is probably because Prime Jet 80 is less penetrated by the hydrophobizate and therefore more condensation products of silanols remain on its surface than for laboratory handsheets. The very similar contact angle of Prime Jet 80 and handsheet samples, despite the apparent large difference in the degree of coating coverage of their surfaces, indicates that the water droplets contact the paper surface in Cassie mode rather than Young’s mode and therefore the average size of the microparticles and their uniform distribution are more important than their chemical composition or mode of bonding to the surface, as it was observed for many superhydrophobic coatings [28].

In the case of the handsheets from subsequent recycling cycles (Figure 8), a small fraction of the spherical organosilicon objects remains permanently bound to the fibers, as they are also visible in the SEM images of the samples before coating. Although there are not enough of them to impart hydrophobic properties to the surface, it explains the decrease in water intake in Cobb 120 s measurements visible in the uncoated set of samples in Figure 5a.

### 3.5. Influence of Coating and Recycling on Mechanical and Optical Properties of Samples

The measured mechanical properties of tested papers are given in Table 4 and Table 5.

The coating seems to have very little impact on mechanical properties. For tensile index and elongation, one can observe a slight increase in values after coating, and for tear resistance, a slight decrease. In both cases, however, the change in the values of these parameters is within the statistical error.

The collected data for recycled handsheets (Table 5) show that in the case of the tensile index and elongation, there is a noticeable increase in these parameters after the first cycle, both for coated and non-coated samples, and then their stabilization in the following cycles. However, although much stronger, for tear resistance the effect is reversed, but also noticeable mainly after the first cycle. For samples from the reference set that were not exposed to the hydrophobizate at all, the first cycle also shows similar trends in all strength parameters, albeit to a more limited extent. In conjunction with previous conclusions from whitewater contamination measurements and SEM images, it can be concluded that these effects are a result of both the leaching of short cellulose fibers and the retention of hydrophobizate components on the fibers, but the latter is more significant.

The desired values of the roughness of the surface and air permeance of the paper depend on its application, hence the hydrophobizing agents should not adversely affect this parameter for a given type of paper. In the case of all paper samples except Prime Jet 80, the surface roughness substantially decreases after coating by around a factor of 2 (see Table 6). Prime Jet 80 has originally very low roughness; therefore, coating had the opposite effect. A large reduction in the AvantLiner roughness value can be a problem in some applications, especially for cardboard boxes, which require a relatively high surface roughness as this facilitates their transport and storage. For all other types of paper, a decrease in roughness is rather a positive effect. For the handsheets, both coated and reference sets show only a slight improvement in this parameter after the first cycle, probably due to the loss of the shortest cellulose fibers.

The coating of the samples notably decreased the air permeance as expected because the hydrophobizate penetrates and partially clogs the pores of the paper. However, despite the high increase in barrier properties towards water, all the coated samples still had relatively high air permeability, indicating that the pores were still partially open, which is particularly important for Optima S 80 sack paper.

For printing papers, brightness is a key factor; therefore, coating mixture should not substantially change this parameter. The ISO brightness of the reference and modified samples for Prime Jet 80 printing paper and handsheets proved that coating has a negligible effect on this parameter (Table 7). However, the decrease in ISO brightness for handsheets in successive cycles from the coated set is very visible, which, in conjunction with the barely noticeable change in this parameter in the reference series, suggests that hydrophobizate contamination rather than changes in the composition and structure of the cellulose fibers is responsible for this effect. The optical properties of AvantLiner and Optima S samples were not examined as brown, unbleached papers have unmeasurable low ISO brightness.

## 4. Conclusions

The application of the aqueous mixture of methyltriethoxysilane and starch to mass-produced paper products by the developed coating method results in a significant increase in hydrophobic properties of their surface expressed by contact angle values (106–112°) and a significant improvement in barrier properties exhibited in decreases in water absorption rates in the Cobb test by 3% to more than 100%, as well as significant changes in water penetration dynamics (PDA) diagrams. The more hydrophilic the starting paper samples were, the greater the resulting barrier and surface effects were. In addition, the coating also significantly increases the durability of the paper in prolonged contact with water, as demonstrated by the Cobb 20 h test results.

The mechanical properties of the tested samples do not change significantly. A slight improvement in the tensile index and a slight deterioration in the tear index are in the range of statistical error. However, the surface roughness substantially decreases after coating by around a factor of 2, except for printing paper, which originally has a very low value of this parameter. It is interesting that the air permeance values, despite being confirmed by SEM studies to have sealed a significant portion of the pores, remain quite high.

Experiments with the recycling of hydrophobizate-coated handsheets made of pure bleached softwood pine kraft pulp show that the recycling itself does not significantly interfere with the disintegration and forming process. The hydrophobic surface and barrier properties remain at the same level in subsequent cycles. There is only a slight improvement in barrier properties in the Cobb test for samples not coated with hydrophobizate, probably because of the accumulation of hydrophobizate degradation products on cellulose fibers. This phenomenon, confirmed by SEM images of analyzed samples, also has some influence on changes in mechanical properties, which are positive for tensile indexes and negative for tear resistance.

FT-IR spectroscopic analysis using the attenuated total reflection (ATR) mode was used to analyze the chemical groups on the surface of the paper sample. Due to the small amounts present on the surface of the fiber, the analysis was based on the spectral differences between the coated and uncoated samples. The strong decrease in the intensity of the bands at 1413 and 872 cm^−1^ after coating suggested that both the grafting of oligosilanols onto cellulose and the intermolecular condensation between adjacent adsorbed –Si–OH groups had been substantially enhanced. Additionally, the appearance of a new diagnostic band at 1270 cm^−1^ (Si–CH_3_) confirmed that the surface of the paper was covered with a network of branched polysiloxanes, often in the form of spheres of irregular diameter.

## Figures and Tables

**Figure 1 materials-14-04977-f001:**
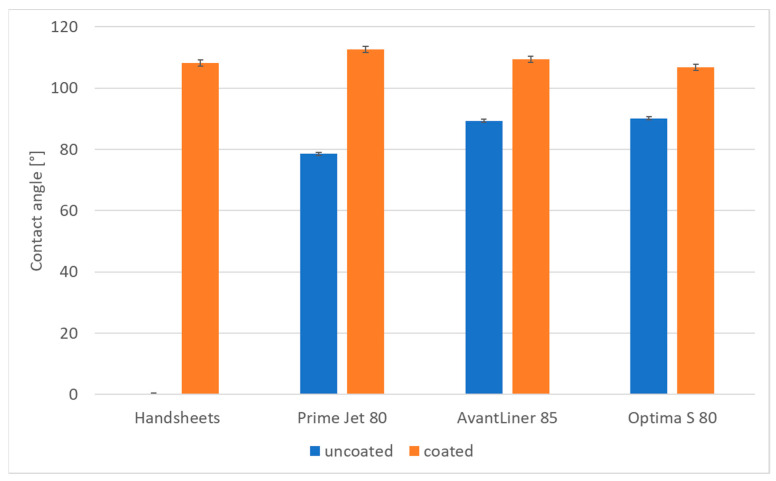
Average water contact angle of uncoated and coated samples of the papers.

**Figure 2 materials-14-04977-f002:**
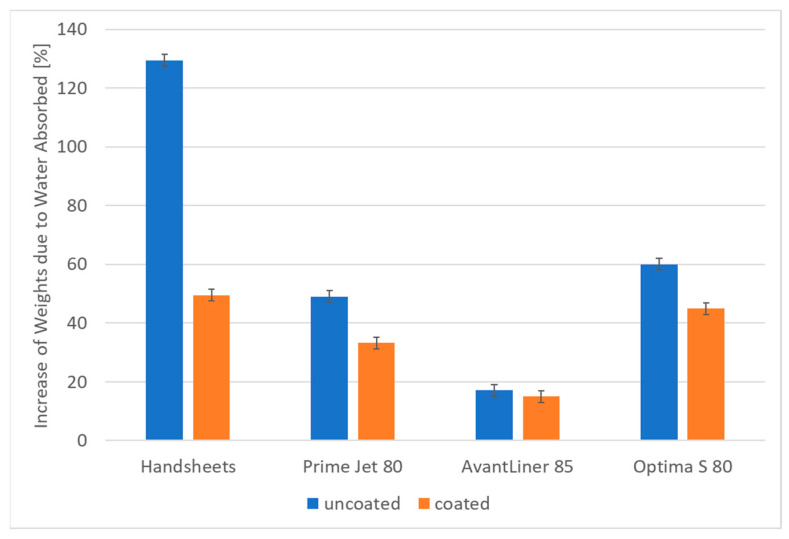
Average increase in weights due to water absorbed (%) (Cobb 120 s) measurements for tested samples.

**Figure 3 materials-14-04977-f003:**
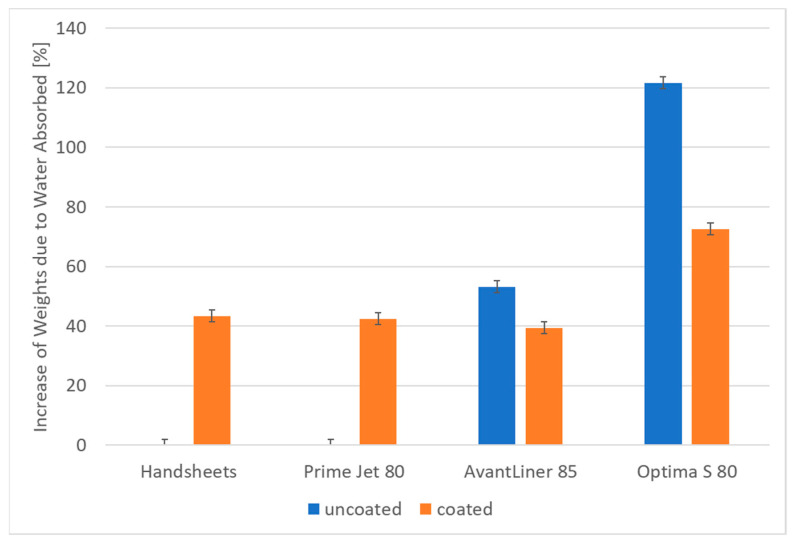
Average increase in weights due to water absorbed (%) (Cobb 20 h) measurements for tested papers.

**Figure 4 materials-14-04977-f004:**
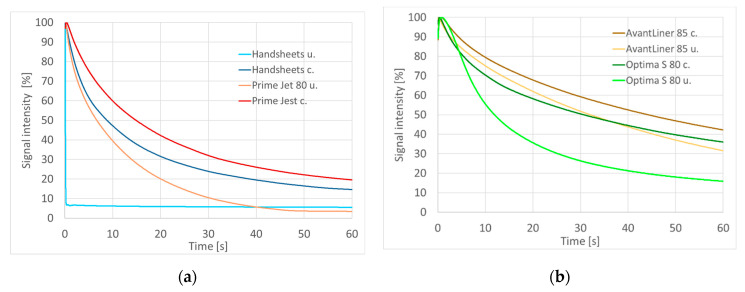
Water penetration dynamics (PDA) plots for (**a**) handsheets and Prime Jet 80, (**b**) AvantLiner 85 and Optima S 80 samples (u.—uncoated, c.—coated).

**Figure 5 materials-14-04977-f005:**
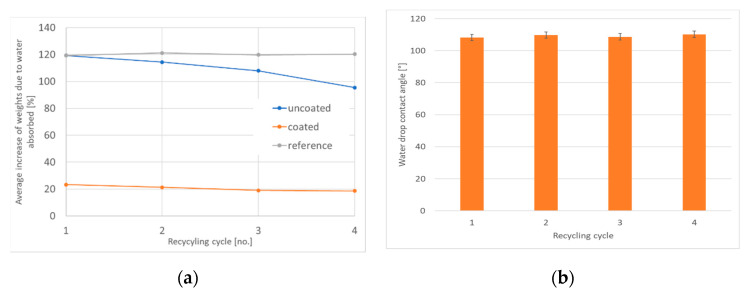
(**a**) Average increase in weights due to water absorbed (%) (Cobb 120 s) measurements for coated and uncoated samples, (**b**) average water contact angles for coated samples formed in 4 repeated cycles.

**Figure 6 materials-14-04977-f006:**
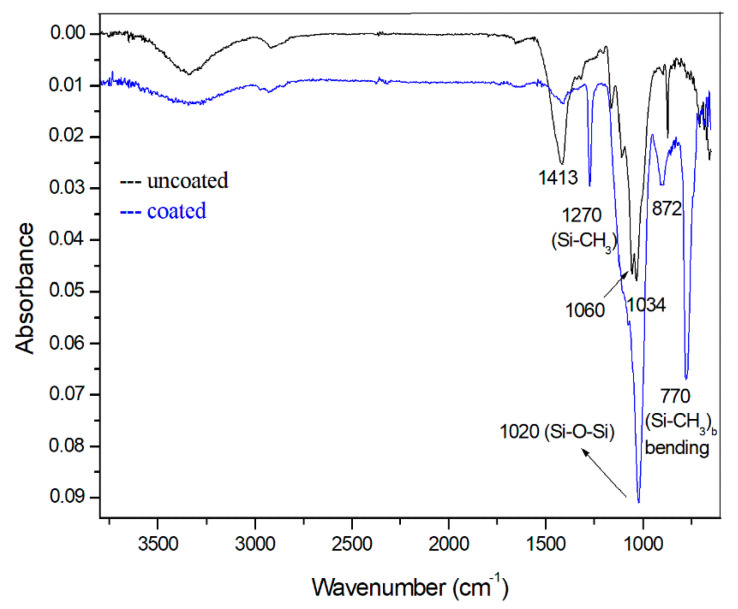
FTIR ATR spectra of uncoated (black line) and coated (blue line) samples of Prime Jet 80 paper.

**Figure 7 materials-14-04977-f007:**
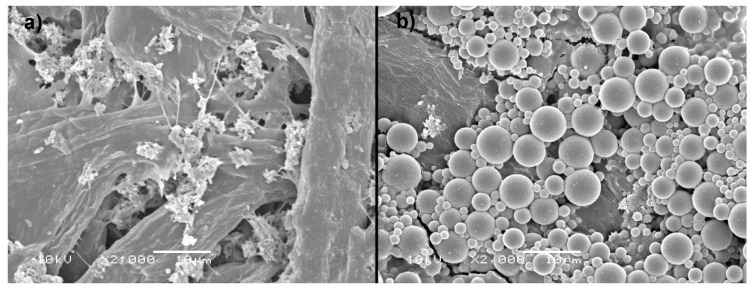
SEM of Prime Jet 80 paper samples: (**a**) uncoated, (**b**) coated.

**Figure 8 materials-14-04977-f008:**
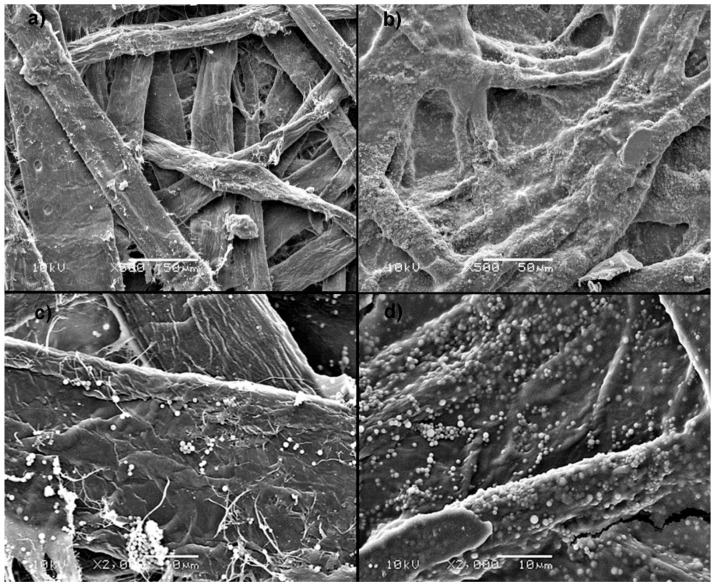
SEM of handsheet samples: (**a**) reference, uncoated, (**b**) first cycle coated, (**c**) 4th cycle uncoated, (**d**) 4th cycle coated.

**Table 1 materials-14-04977-t001:** Arithmetic average weights of paper samples before and after coating.

Sample	Average Weight of Sheets (g)		Increase in Mass (%)
	Before Coating	After Coating	
Prime Jet 80	5.06	5.14	1.58
AvantLiner 85	5.39	5.54	2.78
Optima S 80	5.28	5.35	1.33
Handsheets	2.88	3.01	4.51

**Table 2 materials-14-04977-t002:** Arithmetic average weights of handsheets after each cycle of disintegrating, forming, and coating.

Description	Unit	Cycle 1		Cycle 2		Cycle 3		Cycle 4	
		B^1^	A^1^	B^1^	A^1^	B^1^	A^1^	B^1^	A^1^
Coated set		-	-	-	-	-	-	-	-
Average mass (g)		2.58	2.92	2.64	2.93	2.76	3.04	2.83	3.12
Increase in mass	(g)	-	0.34	-	0.29	-	0.28	-	0.29
	(%)	-	13.2	-	11.0	-	10.1	-	10.2
Reference set	-	-	-	-	-	-	-	-	-
Average mass	(g)	2.58	-	2.54	-	2.56	-	2.58	-

B^1^ = before coating, A^1^ = after coating.

**Table 3 materials-14-04977-t003:** Process water analysis after each cycle.

After:	Cycle 1	Cycle 2	Cycle 3	Cycle 4
Coated set	-	-	-	-
COD (mg/dm^3^)	388	579	734	620
Turbidity (NTU)	10.3	27.0	30.2	29.7
pH	7.01	8.12	8.22	8.16
Reference set	-	-	-	-
COD (mg/dm^3^)	388	285	<25	<25
Turbidity (NTU)	10.3	8.7	<5	<5
pH	7.01	7.03	7.02	7.03

**Table 4 materials-14-04977-t004:** Mechanical properties of tested papers.

Sample	Tensile Index (mN × m^2^/g)		Elongation (%)		Elmendorf Tear Resistance (mN)	
	Uncoated	Coated	Uncoated	Coated	Uncoated	Coated
Handsheets	0.22	0.24	2.08	2.18	349	327
Prime Jet 80	0.28	0.30	2.93	3.13	510	502
AvantLiner 85	0.81	0.79	2.5	2.62	832	812
Optima S 80	0.58	0.61	5.07	5.65	632	618

**Table 5 materials-14-04977-t005:** Mechanical properties of handsheets prepared in recycling cycles.

Handsheet Cycles	Tensile Index (mN × m^2^/g)		Elongation (%)		Elmendorf Tear Resistance (mN)	
	Uncoated	Coated	Uncoated	Coated	Uncoated	Coated
Coated set	-	-	-	-	-	-
1	0.22	0.24	2.08	2.18	349	327
2	0.25	0.26	2.12	2.21	335	274
3	0.26	0.26	2.23	2.53	334	278
4	0.26	0.26	2.23	2.57	345	332
Reference set	-	-	-	-	-	-
1	0.22	-	2.08	-	349	-
2	0.24	-	2.19	-	328	-
3	0.24	-	2.18	-	330	-
4	0.25	-	2.21	-	329	-

**Table 6 materials-14-04977-t006:** Surface roughness and air permeance of paper samples and handsheets prepared in recycling cycles.

Paper	Roughness (mL/min)		Air Permeance (mL/min)	
	Uncoated	Coated	Uncoated	Coated
Prime Jet 80	106	210	153	116
AvantLiner 85	1819	803	163	123
Optima S 80	428	212	469	453
Handsheet coated set cycles				
1	146	132	352	175
2	141	130	320	177
3	143	129	318	175
4	142	131	319	174
Handsheet reference set cycles				
1	146	-	352	-
2	142	-	358	-
3	141	-	346	-
4	143		349	-

**Table 7 materials-14-04977-t007:** ISO brightness for Prime Jet 80 and handsheets.

Paper	ISO Brightness (%)	
	Uncoated	Coated
Prime Jet 80	1.281	1.284
Handsheet coated set cycles		
1	1.237	1.227
2	1.129	1.138
3	1.188	1.215
4	1.070	1.100
Handsheet reference set cycles		
1	1.237	-
2	1.236	-
3	1.235	-
4	1.235	-

## Data Availability

The data are available under an address: https://tulodz-my.sharepoint.com/:f:/g/personal/tomasz_ganicz_p_lodz_pl/Eq4do1oBR5ZJv499MNsByzIBZQYY_sQceeRyIp2yyBgMng?e=CHjzc1, accessed date: 30 July 2021.

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
