# Peer review of "Siloxane-Starch-Based Hydrophobic Coating for Multiple Recyclable Cellulosic Materials"

_materials, 2021, doi:10.3390/ma14174977_

Round 1
Reviewer 1 Report
This study reported a hydrophobic coating which can be effective for mass-produced cellulosic papers. In this manuscript, the water contact angle, water absorptiveness and durability and other performances of uncoated and coated cellulosic samples were carefully investigated and analyzed. But some parts lack of explanation, for example, the formation of silicon particles and the chemical bonding. Although this work may be industrially interesting, I do not believe that enough science is elucidated in the article. Detailed comments are as follows.
1 What is the function of NaOH solution in this study?
2 Does NaOH facilitate the silanol condensation to form silicon particles? In order to understand the formation of silicon particles, silanol condensation experiment without adding NaOH needs to be investigated.
3 In lines 106 and 216, what is the proportion of NaOH in hydrophobic coating? 0.2 wt% or 0.4 wt%?
4 In lines 333 and 334, the peak at 1034 cm-1 demonstrates the formation of covalent bonding. However, this is the uncoated sample of FTIR spectra.
Besides, the demonstration of chemical bonding formation via FTIR analysis is not enough due to some characteristic peaks overlapping with cellulose and starch, XPS analysis needs to be performed to understand the chemical mechanism if necessary.
5 Handsheets show poor resistance to water. Why do you continue to choose handsheets in following recycling experiments?
6 The reference values of ISO Standard should be included in Tables 3 and 7. In this way, we can know whether these test results meet ISO standards or not.
Author Response
1 What is the function of NaOH solution in this study?
The main function of NaOH in the assay is starch glycation. Starch glycation is routinely performed in the paper industry using NaOH or enzymes. The concentration of NaOH used in this work is a typical concentration used industrially for this purpose. The optimum value of this concentration in the final hydrophobizing mixture was selected based on the studies described in our previous publication cited in the current paper (reference 14).
2 Does NaOH facilitate the silanol condensation to form silicon particles? In order to understand the formation of silicon particles, silanol condensation experiment without adding NaOH needs to be investigated.
Yes, pH does affect the condensation of silanols. This has been investigated in two of our previous publications, including an attempt to use a hydrophobizing agent without starch (and therefore also without NaOH), which did not yield good results (See references 14 and 18), as it is also indicated in first paragraph of sec. 3.1 in the present paper. In the present publication, we use the optimal NaOH concentration and starch derived from previous studies, as explained in the article.
3 In lines 106 and 216, what is the proportion of NaOH in hydrophobic coating? 0.2 wt% or 0.4 wt%?
It is 0.2% wt. In line 216 it was typing mistake. Thank you for spotting this.
4 In lines 333 and 334, the peak at 1034 cm-1 demonstrates the formation of covalent bonding. However, this is the uncoated sample of FTIR spectra. Besides, the demonstration of chemical bonding formation via FTIR analysis is not enough due to some characteristic peaks overlapping with cellulose and starch, XPS analysis needs to be performed to understand the chemical mechanism if necessary.
After a closer look at the spectra we have to agree, the signal at 1034 cm-1 is completely covered by the 1020 cm-1 signal from the Si-O-Si bonds, so the IR spectrum proves that the surface is covered by silicon material, but there is no clear evidence of binding to cellulose. Therefore, we have reformulated this section accordingly. (lines 366-376 in sec. 3.4) Considering our previous NMR studies (reference 14) of the nature of these interactions we can conclude that there are probably no direct Si-O-C bonds between silicone particles and cellulose. Unfortunately, we do not have access to XPS.
5 Handsheets show poor resistance to water. Why do you continue to choose handsheets in following recycling experiments?
Exactly for this reason. Due to the poor water resistance of the handsheets, the effectiveness of the hydrophobizate is well seen and not affected by other sizing agents. In the case of recycling tests, carrying out pulping and re-moulding of samples of commercially produced papers - would require this to be carried out on paper machines, which would be associated with enormous costs. For example, AvantLiner paper is produced on a 7.5 m wide machine with a web formation speed of 70-80 km/h. In addition, we preferred to carry out recycling tests on pure cellulosic pulp samples, as the chemical additives used in the production of commercial papers would probably overlap with the effects resulting from the use of hydrophobizate.
6 The reference values of ISO Standard should be included in Tables 3 and 7. In this way, we can know whether these test results meet ISO standards or not.
We do not quite understand what ISO values should be included in these tables. ISO standards specify how to test the given parameters of water and paper, while there are no ISO values for the measured parameters. The acceptable values for these parameters - in the case of water, depend on the production methods used and what the purification equipment allows, and in the case of brightness on the specifications required by the customer. In case of water contamination parameters, we have added the typical, industrial values of them in the discussion (lines 344-349 in sec. 3.3).
Reviewer 2 Report
This manuscript mainly reports siloxane-starch based hydrophobic coating for multiple recyclable cellulosic materials. The key novelty of this work is to realize industrial application of a new hydrophobization agent for mass-produced cellulosic materials. There are several issues that I believe need to be addressed before the publishing of this manuscript.
Major comments:
1. Figure 1 shows the variation of average water contact angle on the tested papers. However, the different wetting states such as absorption, hydrophilicity and hydrophobicity, should be further illustrated in the form of photographs or short videos of contact angle in this manuscript.
2. In the “3.3. part recycling cycles studies”, Fig. 5a presents the average results of water penetration on uncoated, coated and reference samples, respectively. However, Fig. 5b only shows the variation of contact angle on coated samples, which need the data of uncoated and reference samples.
3. The author utilizes two metrics, namely the tensile index and tear index, to measure the mechanical properties of tested samples. However, the author didn’t mention whether the applied coating is wear-resistant, which needs additional experiments.
4. The symbols c) and d) in Figure 8 are separated from corresponding SEM images, which needs modification, and all the SEM images in this figure should be integrated into one page.
5. There are too many tables, such as table 4 to table 7 in this manuscript for comparing the mechanical and optical properties of different samples, which are confusing for the readers. The authors need to summarize theses dates into suitable charts for direct comparison.
Overall, the manuscript is clearly written, and the data presented in this manuscript considers most of the factors that affect the performance. I would recommend for publication if the authors would address all the mentioned problems.
Author Response
- Figure 1 shows the variation of average water contact angle on the tested papers. However, the different wetting states such as absorption, hydrophilicity and hydrophobicity, should be further illustrated in the form of photographs or short videos of contact angle in this manuscript.
We do not think that the photographs of the droplets add anything new to the discussion, because they are quite typical. In the case of handsheet samples the droplets are absorbed very quickly (15 sec.), while in the case of hydrophobic samples the droplets lie steadily on the surface after application and gradually evaporate, as mentioned in the article. It does not appear that any further conclusions can be drawn from the photographs as to the nature of the interactions of water with the samples. We have added several exemplary sets of photographs (one set for all types of papers) of droplets for non-hydrophobic and hydrophobic samples to the supplementary materials:
https://tulodz-my.sharepoint.com/:f:/g/personal/tomasz_ganicz_p_lodz_pl/ErqRS1S_sptNq6F3eTeI654B9Y7maOYRz_EQVfkVoo3ANQ?e=VNLpoX - In the “3.3. part recycling cycles studies”, Fig. 5a presents the average results of water penetration on uncoated, coated and reference samples, respectively. However, Fig. 5b only shows the variation of contact angle on coated samples, which need the data of uncoated and reference samples.
As it is mentioned in the text (lines 311-312) just above the Fig 5., due to the rapid absorption of droplets by handsheets not coated with hydrophobizate, it was not possible to measure static wetting angles for these samples. - The author utilizes two metrics, namely the tensile index and tear index, to measure the mechanical properties of tested samples. However, the author didn’t mention whether the applied coating is wear-resistant, which needs additional experiments.
This is true - we did not measure these values, due to lack of equipment. - The symbols c) and d) in Figure 8 are separated from corresponding SEM images, which needs modification, and all the SEM images in this figure should be integrated into one page.
This was fixed. Thank you for comment. - There are too many tables, such as table 4 to table 7 in this manuscript for comparing the mechanical and optical properties of different samples, which are confusing for the readers. The authors need to summarize theses dates into suitable charts for direct comparison.
In the case of these tables - the differences in many of the values are so small (quite often in the range of statistical error as it is mentioned in publication) that trying to represent them as graphs would not be very informative, moreover in the text we compare many different parameters to each other - so this would require a separate graph for almost every sentence in the sec 3.5, which might prove even less clear than the tables, which we tried to make as clear as possible. In those places where it gave a good effect we used graphs, in the rest tables.
Reviewer 3 Report
In this paper, a siloxane protective film was efficiently formed on the surface of a cellulosic materials to impart hydrophobicity.
This paper is well written and can be accepted, if the authors answer the following questions properly.
- Is it possible to quantitatively show how much the cost will be reduced compared to the conventional method? This includes not only the price of the reagents, but also the total working time from pretreatment to washing after treatment.
- How is the weather resistance of the produced silane protected paper?
- Further discussion is needed on the the environmental load. Please add discussions on the toxicity of reagents required for coating, waste liquid including recycling process, and environmental load when disposing of produced coated paper.
- How durable is the silica coat on both physical and chemical aspect?
Author Response
1. Is it possible to quantitatively show how much the cost will be reduced compared to the conventional method? This includes not only the price of the reagents, but also the total working time from pretreatment to washing after treatment.
In terms of the cost of the agents themselves, the price of methyltriethoxysilane for bulk purchases is in the range of $1-2 per kg (see: https://www.alibaba.com/showroom/methyltriethoxysilane.html ), while the other compounds used in the formulation have prices below $0.5 per kg. Accurate estimation of indirect costs would be difficult at this stage, although it appears that they should not be significant. When dispensing the agent onto typical starch presses, one additional mixer would be needed immediately prior to feeding the agent onto the rollers, which could be operated fully automatically. It would also be possible to apply the agent using standard sprays that are used for e.g. hydrophobizing with AKD. Then also - the only additional cost would be one additional mixer. Such thoughts - due to the fact that we do not have any hard evidence for them - we would prefer not to include in a scientific article.
- How is the weather resistance of the produced silane protected paper?
We have not investigated this, although judging from the results of the "extended Cobb test" one might expect that there would be a significant increase in the resistance of, for example, cardboard boxes to rain.
- Further discussion is needed on the environmental load. Please add discussions on the toxicity of reagents required for coating, waste liquid including recycling process, and environmental load when disposing of produced coated paper.
We added discussion regarding the toxicity of the reagents in the section 3.3 (lines 299-308) . The general environmental load to whitewater had been already discussed but we added values of the typical level of contamination found in industry to better illustrate relatively low effect of using our hydrophobizing agent (lines 344-349).
- How durable is the silica coat on both physical and chemical aspect?
This has not been tested - the agent added does not create a uniform film, but a layer of silica particles, so there is no problem of the film pulling away from the surface or cracking, but the silica particles may possibly pull away because of friction with rough surfaces.
Round 2
Reviewer 1 Report
The authors have addressed my comments
Reviewer 3 Report
I have confirmed all replies and revisions. This paper has been properly revised and is acceptable.